# PPARγ—A Factor Linking Metabolically Unhealthy Obesity with Placental Pathologies

**DOI:** 10.3390/ijms222313167

**Published:** 2021-12-06

**Authors:** Sebastian Kwiatkowski, Anna Kajdy, Katarzyna Stefańska, Magdalena Bednarek-Jędrzejek, Sylwia Dzidek, Piotr Tousty, Małgorzata Sokołowska, Ewa Kwiatkowska

**Affiliations:** 1Clinical Department of Obstetrics and Gynecology, Pomeranian Medical University, 70-204 Szczecin, Poland; kwiatkowskiseba@gmail.com (S.K.); m.bednarekjedrzejek@gmail.com (M.B.-J.); sylwiadzidek@wp.pl (S.D.); piotr.toscik@gmail.com (P.T.); sokolowska.malgorzata@o2.pl (M.S.); 2Department of Reproductive Health, Centre of Postgraduate Medical Education St. Sophie’s Obstetrics and Gynecology Hospital, 01-004 Warsaw, Poland; anna.kajdy@cmkp.edu.pl; 3Department of Obstetrics, Medical University of Gdańsk, 80-952 Gdańsk, Poland; kciach@wp.pl; 4Clinical Department of Nephrology, Transplantology and Internal Medicine, Pomeranian Medical University, 70-204 Szczecin, Poland

**Keywords:** PPARγ, preeclampsia, obesity, metabolically unhealthy obesity

## Abstract

Obesity is a known factor in the development of preeclampsia. This paper links adipose tissue pathologies with aberrant placental development and the resulting preeclampsia. PPARγ, a transcription factor from the ligand-activated nuclear hormone receptor family, appears to be one common aspect of both pathologies. It is the master regulator of adipogenesis in humans. At the same time, its aberrantly low activity has been observed in placental pathologies. Overweight and obesity are very serious health problems worldwide. They have negative effects on the overall mortality rate. Very importantly, they are also conducive to diseases linked to impaired placental development, including preeclampsia. More and more people in Europe are suffering from overweight (35.2%) and obesity (16%) (EUROSTAT 2021 data), some of them young women planning pregnancy. As a result, we will be increasingly encountering obese pregnant women with a considerable risk of placental development disorders, including preeclampsia. An appreciation of the mechanisms shared by these two conditions may assist in their prevention and treatment. Clearly, it should not be forgotten that health education concerning the need for a proper diet and physical activity is of utmost importance here.

## 1. Introduction

Obesity is a known factor in the development of preeclampsia. This paper links adipose tissue pathologies with aberrant placental development and the resulting preeclampsia. PPARγ, a transcription factor from the ligand-activated nuclear hormone receptor family, appears to be one common aspect of both pathologies. It is the master regulator of adipogenesis in humans. At the same time, its aberrantly low activity has been observed in placental pathologies.

## 2. Obesity and Preeclampsia

### 2.1. Definition and Epidemiology of Preeclampsia and Obesity

Obesity, defined as a body mass index (BMI) exceeding 30, is the cause of numerous metabolic disorders and an independent risk factor for death regardless of the cause. According to the WHO, obesity is one of the most serious health problems globally, with the number of incidences tripled over the last 40 years. It is caused by excessive calorie intake and a lack of physical activity. Adipose tissue is an organ that stores backup energy via lipogenesis. In times of nutrient deprivation, fatty acids that are a source of energy are released by way of lipolysis. This mechanism was necessary in the distant past, when our ancestors experienced alternate periods of good access to food and long periods of hunger. Obesity is also a risk factor for the development of preeclampsia [1]. Preeclampsia (PE) is a disease that affects 2–5% of pregnant women and is one of the leading causes of maternal and perinatal morbidity and mortality. Globally, 76,000 women and 500,000 babies die each year from this disorder [2]. The definition of preeclampsia is a matter of continued discussion and is frequently updated. Preeclampsia was initially defined as the presence of hypertension and confirmed proteinuria [3]. In 2013, ACOG expanded the definition by including liver function disorders, hematologic abnormalities, and intrauterine growth restriction [4]. The ISSHP definition published in 2018 incorporated cases with aberrant flows in the umbilical arteries and cases affected by intrauterine fetal death [5]. More and more well-respected authors are now allowing for the possibility of preeclampsia presence where there is no accompanying hypertension, a factor that was previously considered a sine qua non for such diagnosis. Numerous reports suggest that PE pathogenesis should only be classified based on the timing of its development secondary to ischemic placental dysfunction [6,7]. The former type of PE mainly develops due to deficient invasion of the uterine wall by placental trophoblast cells. The latter is impaired primarily due to premature aging of the placenta and maternal predispositions [8]. The angiogenesis markers sFlt-1 and PlGF have become well established in diagnosing ischemic syndromes [9,10,11]. Strong correlations between BMI and the risk of developing preeclampsia have been identified [12]. This risk doubles at a BMI of 26 kg/m^2^ and triples at 30 kg/m^2^ [13]. Body weight reduction has been found to reduce that risk. In the search for mechanisms responsible for the higher incidence of preeclampsia in obese pregnant patients, such factors have been identified as metabolic factors, compounds produced by the adipose tissue, and imbalances between vasodilating and vasoconstrictive agents.

### 2.2. Healthy and Unhealthy Obesity

The adipose tissue is an energy storage area, and it is also responsible for the production of multiple cytokines. Excess adipose tissue may be metabolically healthy or unhealthy. Healthy fat tissue is the one that is deposited subcutaneously (subcutaneous fat tissue (SAT)), which is insulin sensitive, ensuring that glucose levels are maintained at all times, and inhibits hepatic gluconeogenesis. Unhealthy fat tissue is the one that is deposited viscerally (visceral adipose tissue (VAT)) in the liver and in the skeletal muscle (ectopic adipose tissue (EAT)), which is insulin resistant, and produces cytokines responsible for local and systemic inflammation [14,15,16]. Inflammatory cytokines TNF-α, IL-6, and leptin secreted by the VAT activate endothelial cells by activating leukocyte adhesion proteins, leading to tissue infiltration by inflammatory cells—mainly macrophages and monocytes—which, by producing further inflammatory cytokines, induce an inflammatory loop [17,18]. Adiponectin, present in low concentrations in obese persons, is a favorable anti-inflammatory cytokine secreted by the adipose tissue.

### 2.3. How Obesity Predispose to Preeclampsia

It is well known that inflammation and endothelial dysfunction impair trophoblast proliferation, migration, and invasion and thus contribute to the development of preeclampsia [1,19,20]. Nitric oxide in our systems is formed from L-arginine using three nitric oxide synthases: endothelial (eNOS/NOS3), inducible (iNOS/NOS2), and neuronal (nNOS/NOS1). NO produced by eNOS is responsible for vasodilation, increasing vascular blood flow, and inhibiting platelet aggregation and the proliferation of smooth muscle cells [21,22,23]. NO produced by eNOS also inhibits the endothelial production of adhesion proteins, thus inhibiting the infiltration of tissues by inflammatory cells. In metabolically unhealthy obesity, NO production by eNOS is significantly reduced, while iNOS production by adipocytes and macrophages is significantly elevated [24,25,26]. At the same time, damaged endothelial cells produce endothelin-1, a potent vasoconstrictor. The imbalance between eNOS and endothelin causes vasoconstriction, increased peripheral resistance, and arterial hypertension [1]. Some authors have confirmed lower eNOS concentrations in preeclamptic patients [27,28]. Such patients have also been observed to have higher endothelin-1 concentrations and a positive correlation between endothelin-1 concentrations and the severity of preeclampsia [29,30]. Reactive oxygen species (ROS), whose concentrations in obesity and hyperglycemia grow significantly, are also an issue. ROS are produced by NADPH oxidase, via mitochondrial oxidative phosphorylation, and by endothelial cells [31,32]. The inflammatory infiltration increases ROS production by its macrophages. ROS, in turn, exacerbate inflammation and endothelial dysfunction. We know what disorders are present in obesity and observe similar disorders in preeclampsia, but we do not know the mechanisms behind these disorders.

## 3. Obesity

According to multiple epidemiological studies, 10–40% of obese people are metabolically healthy. The metabolically healthy obese (MHO) phenotype demonstrates good sensitivity to insulin, normal arterial pressure, and normal lipid, inflammatory, hepatic, and hormonal profiles [33]. On the other hand, metabolic disorders may occur in persons with normal body weight. Despite their normal body weight, such patients have hyperinsulinemia, show insulin resistance, an increased risk of type 2 diabetes, hypertriglyceridemia, and atherosclerosis [34]. There have been many attempts at explaining this situation.

### 3.1. Body Composition

One of them focuses on body composition. The NHANES III study, carried out in the USA, revealed that 10% of the population with a normal BMI had an increased fat content [35,36].

### 3.2. Adipose Tissue Distribution

Another explanation points to adipose tissue distribution. In terms of its location in the body, adipose tissue is either truncal or peripheral. Truncal fat is found in the subcutaneous tissue of the chest and abdomen and the fatty tissue present inside the chest and the abdomen. Peripheral fat is that associated with the subcutaneous fat of the lower and upper extremities. The first-ever observation of correlations between fat distribution and cardiovascular diseases was recorded by Vague in 1947, who concluded that, when compared with gynoid obesity, android obesity was associated with a higher incidence of diabetes mellitus, ischemic heart disease, gout, and nephrolithiasis [37]. Similar findings were made for the waist-to-hip ratio. A high value of the ratio was associated with an increased risk of stroke, infarction, or death [38]. INTERHEART, a very large observational study carried out in 52 countries, confirmed that the waist-to-hip ratio was an independent risk factor for myocardial infarction. The visceral fat tissue is popularly considered to have the most pronounced effect on metabolic disorders. According to the portal vein theory, free fatty acids produced via VAT lipolysis enter the liver via the portal vein, thus leading to lipid synthesis, gluconeogenesis, and insulin resistance. The fact that the VAT only accounts for 15–18% and 7–8% of the total adipose tissue in men and women, respectively, makes this theory controversial. Free fatty acids derived from the VAT only account for 15% of total free fatty acids [39,40,41]. There are, therefore, doubts as to whether the VAT is in fact the only determinant of insulin resistance. Abate et al. conducted a study in non-diabetic men with different levels of obesity. They concluded that the subcutaneous adipose tissue in the truncal region was of greater clinical significance than visceral and extraperitoneal fat depots [41]. The same group of researchers studied men with insulin-independent type 2 diabetes and found them to have subcutaneous rather than peripheral and visceral adipose tissue [42]. Many researchers have confirmed the relationship between subcutaneous fat and insulin resistance and atherosclerotic lesions [43,44]. In men, truncal subcutaneous fat is 4–5 times the volume of visceral fat. Similarly, in women, the volume of abdominal subcutaneous fat (the L4-L5 region) is five times larger than that of visceral tissue at this level [40,41,45]. Due to its volume, the truncal subcutaneous adipose tissue appears to be the major source of free fatty acids and insulin resistance.

### 3.3. Inflammatory Infiltration of Adipose Tissues

Multiple studies have found that adipose tissue with inflammatory infiltration of macrophages is the main contributor to insulin resistance and endothelial dysfunction. Attempts to find the factor causing inflammation have led to new discoveries.

## 4. Adipogenesis and PPARγ Activity

### 4.1. Hypertrophy and Hyperplasia of Adipose Tissue

The expansion of adipose tissue can be driven either by increasing the sizes of the existing adipocytes (hypertrophy) or by forming new ones (hyperplasia/adipogenesis). Adipogenesis is achieved by differentiating mesenchymal cells into preadipocytes and then adipocytes [46]. Adipogenesis is not yet fully understood, although BMP4 (bone morphogenetic protein-4) is known to participate by activating zinc-finger protein 521 (ZNF52), which, in turn, activates PPARγ.

### 4.2. Physiology of PPARγ

Peroxisome proliferator-activated receptor-γ (PPARγ or PPARG), also known as the glitazone reverse insulin resistance receptor, or nuclear receptor subfamily 1, group C, member 3 (NR1C3), is a type II nuclear receptor (a gene-regulating protein) that in humans is encoded by the PPARγ gene [47]. PPARγ is a transcription factor from the ligand-activated nuclear hormone receptor family. PPARγ1 is a common isoform mainly found on macrophages, in the liver, colon, endothelium, kidneys, and placenta. PPARγ2 is only present in adipocytes, while PPARγ3 appears in the adipose tissue, macrophages, and colon epithelium [48]. PPARγ consists of three domains. The N-terminal domain, subject to phosphorylation and dephosphorylation, is responsible for receptor regulation. The ligand-binding domain (LBD) has a complex secondary structure that allows for the binding of different ligands, even as large as fatty acids. A ligand, after binding with PPARγ, changes the structure of the LBD, specifically a part thereof known as activation function 2 (AF2). PPARγ then binds with another transcription factor known as the retinoid X receptor (RXR). Only now will the resulting heterodimer constitute a functional transcription factor. The third domain, known as the DNA-binding domain, binds with specific nucleotide sequences in target gene promoters. It binds with DNA using zinc fingers—this coupling regulates the transcription of target genes by stimulating or inhibiting their expression. Co-activators and co-inhibitors may be involved in this reaction. PPARγ activation has a very broad effect across the human system. Above all, it exerts a very strong anti-inflammatory effect, sensitizes tissues to insulin, and affects the biology of adipocytes, as noted earlier. Patients with PPARγ gene mutations develop lipodystrophy and severe metabolic disorders such as insulin resistance, type 2 diabetes, dyslipidemia, and arterial hypertension [49]. They are involved in cell proliferation and differentiation. Most studies on the role of PPAR have been on animal models since as many as 80% of the amino acids in this protein are identical between species [50]. In times of excess calorie intake, adipocytes respond in two ways. Those with impaired adipogenesis become hypertrophic. Previously, the number of adipocytes was believed to be determined early in life and remain mostly stable through adulthood. It has now been found that there is a relationship between adipocyte size and the system’s capacity to form new adipocytes. People with large subcutaneous adipocytes have little ability to differentiate mesenchymal cells into adipocytes, which is due to their impaired PPARγ activation [51,52]. Mature, healthy adipocytes release BMP4 during adipogenesis, which causes mesenchymal cells to differentiate into adipocytes. By way of compensation, hypertrophied adipocytes release an abundance of BMP4, which, however, does not lead to preadipocyte recruitment, as such patients have very high contents of gremlin-1, a BMP4 antagonist [53]. Nevertheless, a prolonged supply of excess calories has been shown to be able to cause the formation of new adipocytes from preadipocytes. By nature, preadipocytes, located around vessels, are similar to fibroblasts. The balance between hypertrophic adipocytes and adipogenic potential has proven to be a major factor for metabolic health. Enlarged adipocytes have been shown to be the main cause of insulin resistance [54,55]. Small adipocytes counteract the metabolic disorders associated with obesity [56,57,58]. Expansion in size exposes adipocytes to mechanical stress due to increased contact with neighboring fat cells thus leading to hypoxia, while their further expansion makes them exceed the limits of oxygen diffusion. The increased mechanical and ischemic stress triggers inflammation [59]. Enlarged adipocytes display more pronounced lipolysis and secrete more inflammatory adipokines and less anti-inflammatory adipokines, such as adiponectin [60,61].

### 4.3. Mechanism of Adipose Tissue Hyperplasia

Adipogenesis begins with a fibroblast-like progenitor cell, which expresses platelet-derived growth factor receptors alpha and beta (PDGFR-α and PDGFR-β) and is restricted to the adipocyte lineage without any morphological changes to form a preadipocyte. This means that, from now on, it will not form other mesenchymal cells such as myoblasts, chondroblasts, or myeloblasts [62,63]. At this stage, BMP 2 and 4 are responsible for promoting the multipotent cell’s adipogenic commitment [64,65]. BMP binds to its receptor, thus stimulating the transcription factor SMAD4 by activating its heterodimeric partners SMAD 1, 5, and 8 [66]. Zinc-finger transcription protein 423 (ZFP423) has been shown to act to sensitize the multipotent cell to BMP signaling [67]. The second step involves growth arrest with lipid accumulation and cell differentiation into mature insulin-sensitive adipocytes. This is affected by SMAD4, which stimulates the transcription of PPARγ, the major regulator of adipogenesis. Another transcription factor, GATA 2 and 3, suppresses adipogenesis by inhibiting PPARγ transcription [68]. PPARγ is the major adipogenic factor, in both in vivo and in vitro adipocyte cultures. Unfortunately, no endogenous ligand for PPARγ is known [69]. PPARγ activates the transcription factor C/EBPα, which is necessary for a preadipocyte to differentiate into an adipocyte. As PPARγ is required for this to occur, a synergistic effect of the insulin receptor and PPARγ/C/EBPα is implicated, leading to the expression of genes typical of mature adipocytes [50] (Figure 1).

## 5. Placental Development

Since preeclampsia is associated with abnormal placental development, understanding this highly complex process is crucial. The placenta begins to develop in the first days upon fertilization. It consists of the prelacunar, lacunar, primary villous, and villous stages.

### 5.1. Prelacunar Stage

In the initial stage, at 5 to 8 days after fertilization, the first cell lineages are differentiated, as a result of which the trophoblast is formed, and the blastocyst begins to develop. The trophoblast cells encircle the embryo creating the so-called trophectoderm. At approx. 6 or 7 days, the blastocyst (mainly its trophectodermal part) attaches to and is subsequently implanted in the endometrium. Only those trophoblast cells that have direct contact with the embryo attach to the uterine wall. During the implantation, the trophoblast proliferates and becomes layered, with its outer cells merging, attaching to the endometrium, and penetrating it, thus resulting in the formation of the first invasive multinucleated syncytiotrophoblast. It is the only embryonic tissue in contact with maternal tissues. The single trophoblast cells hidden behind the invasive syncytiotrophoblast are referred to as the cytotrophoblast. The syncytiotrophoblast resembles invasive finger-shaped processes that reach out to the uterine tissue. It loses its ability to divide but is fused with the cytotrophoblast, which proliferates and offers a continuous supply of nuclei, organelles, enzyme systems, and mRNA transcripts into the syncytiotrophoblast. The cytotrophoblast is the major source of fresh material for the above-lying syncytiotrophoblast.

### 5.2. Lacunar Stage

As pregnancy continues, the number of cytotrophoblast cells decreases significantly, although their ability to proliferate is not diminished. In the subsequent lacunar period, at approx. 8–13 days, the syncytiotrophoblast starts to grow around the embryo to assume a circular shape at the end of this phase. At 8 days, the first vacuoles are formed in the syncytiotrophoblast and join to make larger lacunae that are separated from each other by structures of the syncytiotrophoblast. They extend from the embryonic side to the maternal side of the developing placenta and constitute intervillous spaces that will be filled with maternal blood.

### 5.3. Primary Villous Stage

In the primary villous period, between 13 and 28 days after fertilization, primary villi are formed from the syncytiotrophoblast and the underlying cytotrophoblast. Mesenchymal cells appear behind the cytotrophoblast cells. Secondary villi are formed that consist of mesenchymal cells surrounded by a layer of the syncytiotrophoblast and the cytotrophoblast. At 18–29 days, vasculogenesis in the mesenchyme of the secondary villi begins to produce tertiary villi. Angiogenesis is greatly increased by the hypoxia on this side of the placenta. At this stage, the cytotrophoblast begins to proliferate, causing high pressure and thus forcing the cells to migrate through the syncytiotrophoblast to the maternal side of the placenta to form the extravillous trophoblast that invades the maternal structures.

### 5.4. Extravillous Trophoblast

The first extravillous trophoblast is referred to as the interstitial trophoblast. It develops at approx. 15 days after fertilization. It has the HLA g histocompatibility antigen on its surface, which distinguishes it from the villous trophoblast. After penetrating through the syncytiotrophoblast in the early villous phase, trophoblasts form multiple cellular layers, referred to as trophoblast cell columns. Within the surface that invades the maternal tissue, these cells do not proliferate, as they are merely pushed in by the underlying proliferating extravillous trophoblast cells. They have the ability to invade, similarly to cancer cells, but lose their ability to proliferate, so that upon entering maternal circulation, they will not spread in other tissues. Extravillous trophoblast invasion is not limited to early pregnancy and continues throughout gestation. The cells that first enter the connective tissue of the uterus are termed interstitial extravillous trophoblast. They penetrate one-third of the inner uterine layer. This type of trophoblast anchors the placenta to the uterine wall. The endoglandular trophoblast projects from the interstitial trophoblast and penetrates maternal uterine glands early into pregnancy. It opens up these glands and connects them with the placental intervillous spaces, thus forming a path for the supply of nutrients to the embryo. A specific population of interstitial trophoblast cells reach the uterine spiral arteries, penetrate their walls and endothelium and enter their lumen, transforming them into wide tubes which are now beyond maternal vascular control. The resulting tissue is referred to as the endovascular trophoblast. The endovascular trophoblast provides the interface between the intervillous space and maternal circulation.

### 5.5. Villous Stage

During the last villous stage that beings at 28 days after fertilization and lasts throughout pregnancy, the placenta increases in size by creating syncytial sprouts, or protrusions, underneath which first cytotrophoblasts and then mesenchymal cells and vessels proliferate. A type of villous tree is formed, which spreads throughout pregnancy, although it does so at a slightly lower pace [70].

## 6. PPARγ in Normal vs. Pathological Pregnancy

As we mentioned earlier, overweight and obese women are at a higher risk of developing preeclampsia. Preeclampsia and obesity are accompanied by common disorders such as hyperinsulinemia, insulin resistance, high leptin concentrations, low adiponectin concentrations, high levels of inflammatory cytokines TNFα and IL-6, and lipid profile disorders [1]. Since not all overweight and obese pregnant women have preeclampsia or demonstrate the aforementioned disorders, they appear to demonstrate the so-called metabolically healthy obesity. Therefore, PPARγ appears as a likely common element shared by metabolically unhealthy obesity and placental development disorders.

### 6.1. Differentiation of Progenitor Cells into Trophoblast Cells

As we mentioned earlier, the main placental cells are the villous cytotrophoblast cells that are in fusion with syncytiotrophoblast. They proliferate intensively throughout pregnancy and ensure a continuous supply of nuclei, organelles, enzyme systems, and mRNA transcripts into the syncytiotrophoblast, thus providing fresh material for the syncytiotrophoblast lying above. The syncytiotrophoblast is a cellular layer that lines the placenta, including the villi. Ensuring the differentiation of the trophoblast cells from their earliest progenitor forms is the responsibility of a placental transcription factor termed glial cell missing-1 (GCM1). Its reduced activity in the placentae of PE patients has been proven [71,72]. PPARγ regulates the expression of GCM1, thereby affecting the differentiation of the trophoblasts from the first days of embryo formation.

### 6.2. Trophoblast Cells Proliferation and Formation of Cytotrophoblast and Syncytiotrophoblast Cells

In mice, PPARγ is of key importance in the regulation of the trophoblast as early as 10 days after fertilization. In rats, PPARγ expression is observed at 11 days (which corresponds to 9 days in mice), peaking at 13 and decreasing at 15 days [73]. Mouse fetuses knocked out for the PPARγ gene died at 9 or 10 days, which was due to necrosis [74]. In the same experiment, it was found that the knockout mice demonstrate numerous vascular anomalies on both the fetal and maternal sides of the placenta. The fetal vessels were almost undeveloped and most of the villi showed no trace of them, while the maternal sinuses were enlarged and torn apart, sometimes forming a single reservoir. In addition, maternal erythrocytes which should be found in the sinuses were discovered in cells of the connecting zone of the placenta, thus manifesting phagocytic activity of the trophoblast. In heterogeneous PPARγ ± mice, lesions within the maternal part of the placenta were also present but were less severe [74]. The villus of a healthy placenta has a characteristic three-layer barrier between the maternal sinuses and the fetal endothelium. In addition, the barrier is in close contact with the endothelium of the fetal part of the placenta. In PPARγ−/− mice, the villus is surrounded by an excessively thick trophoblast layer that does not resemble the three-layer barrier of a healthy placenta. Additionally, the contact with the endothelium is less tight [74]. In humans, the natural endogenous activators of PPARγ (fatty acids and lipid metabolism products) are elevated in normal pregnancies (almost by a factor of two), suggesting that its activity is increased under healthy conditions [75]. In the human placenta, PPARγ is mainly found in the trophoblast and is associated with its proliferation as well as maturation [76,77,78]. In the first trimester, it is mainly present in the villous cytotrophoblast, appearing in the extravillous trophoblast at 7 weeks’ gestation [79]. In the second trimester, it is present in the anchoring cytotrophoblast [80].

### 6.3. Extravillous Trophoblast

Recent studies have shown that its expression is downregulated in the syncytiotrophoblast of diabetic patients. In her most recent study of the placentae of diabetic patients, Knabl observed decreased PPARγ expression in both the syncytiotrophoblast and the extravillous trophoblast. This proves that PPARγ is involved in building the site for maternal–fetal exchange but also in the extravillous invasion that is so important for normal blood flow from the mother to the placenta [81].

### 6.4. sFlt-1 Synthesis

Another study, carried out by Armistead, concluded that PPARγ expression was reduced in preeclamptic placentae and involved increased production of soluble fms-like tyrosine kinase1 (sFlt-1). The activation of PPARγ caused decreased sFlt-1expression in the placenta in the first trimester. The same author found a correlation between the expression rates of PPARγ and GCM1. A low GCM1 expression was associated with high sFlt-1 content. It was suggested that the PPARγ/GCM1 axis plays a crucial role in regulating sFlt-1 synthesis [82]. A study, performed by Levytska et al. on a choriocarcinoma cell line, confirmed that increased activity of PPARγ is linked to increased activity of the transcription factor GCM1 [83]. Others have examined vascular endothelial growth factor (VEGF), which in the placenta is responsible for stimulating the formation of blood vessels the development of villous and extravillous trophoblasts. They pointed to a regulatory link between VEGF synthesis and PPARγ activity [84,85]. Wait observed that plasma from mothers with physiological pregnancies stimulates PPARγ, which may suggest the presence of natural endogenous triggers [80].

### 6.5. Transfer of Free Fatty Acids from the Mother to the Fetus

Another problem is that of the transfer of free fatty acids from the mother to the fetus, which is essential for the accurate development of the latter. The role of PPARγ in lipid storage is known. The placenta of animals knocked out for the PPARγ gene had less of the so-called lipid droplets in the villous trophoblast that may play a role in energy storage [74]. One researcher has stated that PPARγ influences free fatty acid uptake by the trophoblast by increasing the expression of fatty acid transporters [86]. These results were confirmed for the human placenta, whereby it was found that PPARγ increases free fatty acid uptake and accumulation by upregulating the expression of a free fatty acid-binding protein [87]. In turn, oxidized LDLs inhibit trophoblast invasion by regulating PPARγ. PPARγ activity appears to be regulated by lipid metabolism. PPARγ may be a nutrition sensor regulating free fatty acid uptake and trophoblast growth and differentiation [88].

### 6.6. Summary of Placenta Disorders Caused by Low PPARγ Activity

Studies of both animal and human placentae have shown that PPARγ is required for the proper development of the placenta from as early as the first days after fertilization. It contributes to the differentiation of progenitor cells into trophoblast cells and to their proliferation, as well as the formation of a placental tree made of cytotrophoblast and syncytiotrophoblast cells. Its activity has been shown to take place in the extravillous trophoblast, and its important role in the proper formation of placental vessels and maternal sinuses has been proven, as well. Another important aspect is its effect on the expression of sFlt mRNA and the synthesis of this agent, which is a marker for impaired placental development, on the one hand, and a factor causing generalized damage to the maternal side of the placenta, on the other. PPARγ is also responsible for ensuring proper transportation of fatty acids from the mother to the fetus and for their storage.

### 6.7. Factors Affecting PPARγ Activity—Epigenetic Changes

PPARγ is a common factor shared by metabolically unhealthy obesity and impaired placental development in obese patients or in non-overweight and non-obese patients with impaired fat metabolism. The most important question is about the factors affecting PPARγ activity: Could that be dysfunctional lipid metabolism, or perhaps a long-term intake of excess calories leading to its permanently low activity secondary to some epigenetic changes? The epigenetic events that affect gene expression include microRNA activity and DNA methylation. Motawi et al. found that obese patients, as well as obese patients with colorectal cancer, had reduced PPARγ activity, which correlated with the contents of microRNAs that are elevated in obesity—namely, miRNA 27b, 130b, and 138. In addition, methylation of the promoter region of the PPARγ gene was associated with its reduced activity and lower plasma concentrations [89]. This experiment proved that obesity can permanently affect PPARγ activity, a disorder that exacerbates inflammation and hypoxia, which, in turn, stimulate epigenetic changes. Thus, a chain reaction is set in motion, which we can attempt to stop using natural and synthetic PPARγ ligands (Figure 2).

### 6.8. Animal Models of Human Placentation—Limitations

Mouse models are often criticized for pregnancy research, as gestation is short, with much of organ development completed postnatally. These models have informed us about the earliest stages of embryonic development. Gene expression patterns in mouse and human placenta best agree when the comparison is restricted to the first 16 weeks of human pregnancy. In the human placenta and most of the other models such as rodents, fetal–maternal interfaces are similar—no maternal tissues are present in the interhaemal barrier. The number of trophoblast layers varies from three to one in different rodents. A further layer of complexity is added by the arrangement of the two bloodstreams. In rodents and many other mammals, the exchange is between fetal and maternal capillaries. However, in the human placenta, maternal blood perfuses the intervillous space. Despite this focus on the interhaemal barrier, there are important differences in the degree to which trophoblast cells invade the underlying endometrium and its blood vessels. No animal model is fully suited to explore this aspect of human placentation. To learn more about the differences in the structure of the human and the animals’ placenta used in research studies cited in this article, I refer readers to the article by Anthony M. Carter entitled “Animal models of pregnancy and human location: alternatives to mice” [90].

## 7. PPARγ as a Therapeutic Option in Assisting Proper Placental Development

PPARγ activity can be used as a therapeutic option in the treatment and prevention of placental disorders.

### 7.1. PPARγ Agonist Rosiglitazone Andpioglitazone

McCarthy concluded that pregnant mice administered with PPARγ antagonists developed a phenotype similar to that in human preeclampsia [91]. In another experiment, the same author proved that the administration of rosiglitazone, a synthetic PPARγ agonist, to rats with impaired placental flow greatly improved the parameters of that flow [91]. Studies on animal models have not implicated any adverse effects of rosiglitazone on the fetus. There have been reports of pregnant women receiving rosiglitazone or pioglitazone where no adverse effects on the fetus were observed, even though the agents penetrated the placental barrier [92,93,94]. In those studies, the administered doses were below the recommended values in the treatment of diabetes, and the administration time was between 7 and 17 weeks of gestation. The latter of these drugs, pioglitazone, is registered for the treatment of type 2 diabetes.

### 7.2. Essential Fatty Acids

The activity of PPARγ is controlled by endogenous ligands produced via fatty acid metabolism; hence, the compound is referred to as a lipid sensor. The region of PPARγ that binds ligands is structured so that it is able to bind even very large molecules, a unique feature among nuclear receptors. The ligand-binding part binds essential fatty acids (EFAs) that act as PPARγ agonists [95,96,97,98] Acids that act as agonists include docosahexaenoic acid (DHA), eicosapentaenoic acid (EPA), and α-linolenic acid (ALA), which are omega-3 polyunsaturated fatty acids. In an experiment on monocytes stimulated to secrete IL-1, IL-6, and TNF, the addition of ALA reduced their secreted amounts. The author proved that the stimulation of PPARγ was behind this effect [99,100]. Another author also proved the stimulating effects of DHA and EPA on PPARγ [100,101]. Linolenic acid (LA) and its metabolites are well-known PPARγ agonists [102]. Eicosanoids, which are products of the metabolism of omega-6 acids, mainly arachidonic acid, have stimulating effects on PPARγ, as well. These include leukotriene B4 (LTB4) and prostaglandin J2 (PGJ2) [97]. To activate PPARγ, both EFAs and eicosanoids must come in high concentrations (approx. 100 uM) [98]. Phytanic acid, a saturated branched long-chain fatty acid, is also a natural PPARγ agonist [102]. Synthetic ligands also include fibrates, which are medicines used to treat hypertriglyceridemia [103,104,105,106,107].

### 7.3. Glucocorticosteroids and Circadian Rhythm

Glucocorticosteroids are also PPARγ stimulants, although only when the physiological rise in glucocorticoids in the morning is maintained, as followed by their drop in the evening and during the night. They activate the transcription factors PPARγ and C/EBPs and sensitize preadipocytes to insulin [108]. The obese maintain consistently elevated concentrations of glucocorticosteroids. A disruption of the circadian rhythm poses a higher risk of obesity and metabolic syndrome. The circadian rhythm is necessary for the correct activation of PPARγ and for adipogenesis to occur. This may be due to the absence of a peak in the glucocorticosteroid concentration that occurs in the morning after a night’s rest. Tests in which dexamethasone was administered have confirmed this finding. Short-term administration stimulates adipogenesis, while a longer administration exceeding 123 h inhibits the process [109], hence the importance of maintaining a proper circadian rhythm by pregnant patients.

### 7.4. Acetylsalicylic Acid

According to the FIGO 2019 recommendations, all women at risk of developing preeclampsia, among them obese patients, should take 150 mg of acetylsalicylic acid daily. Such prevention significantly reduces the risk of PE [2]. In a study by Zhang et al., PE was induced in rats, after which the study group received aspirin. Subsequently, the placentae were examined for PPARγ activity and contents. In a group that did not receive aspirin, the concentration and activity of PPARγ were low. The group that received aspirin demonstrated much higher PPARγ concentration and activity. This experiment proved that such a significant drop in the likelihood of developing PE in patients on aspirin may be due to elevated PPARγ contents [110].

## 8. Conclusions

Overweight and obesity are very serious health problems worldwide. They have negative effects on the overall mortality rate. Very importantly, they are also conducive to diseases linked to impaired placental development, including preeclampsia. More and more people in Europe are suffering from overweight (35.2%) and obesity (16%) (EUROSTAT 2021 data), some of them young women planning pregnancy [111]. As a result, we will be increasingly encountering obese pregnant women with a considerable risk of placental development disorders, including preeclampsia. An understanding of the mechanisms shared by these two conditions may assist in their prevention and treatment. Clearly, it should be borne in mind that health education concerning the need for a proper diet and physical activity is of utmost importance here.

## Figures and Tables

**Figure 1 ijms-22-13167-f001:**
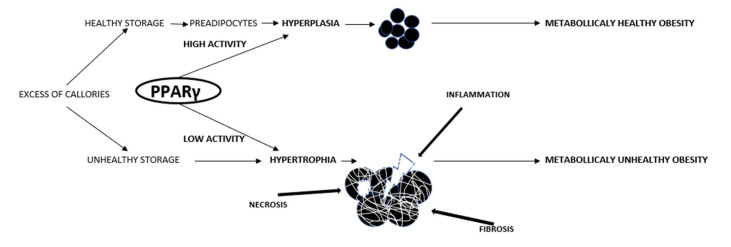
The mechanisms of adipose tissue expansion. In the state of overnutrition, the expansion of adipose tissue can be driven either by increasing the sizes of the existing adipocytes (hypertrophy) or by forming new ones (hyperplasia/adipogenesis). Adipogenesis is achieved by differentiating mesenchymal cells into preadipocytes and then adipocytes. Adipogenesis begins with fibroblast-like progenitor cells (kind of mesenchymal cells) and is restricted to the adipocyte lineage without any morphological changes to form a preadipocyte. This means that from now on it will not form other mesenchymal cells such as myoblasts, chondroblasts, or myeloblasts. At this stage, BMP 2 and 4 are responsible for promoting the multipotent cell’s adipogenic commitment. The second step involves growth arrest with lipid accumulation and cell differentiation into mature insulin-sensitive adipocytes. This is affected by SMAD4, which stimulates the transcription of PPARγ, the major regulator of adipogenesis. Expansion in size exposes adipocytes to mechanical stress due to increased contact with neighboring fat cells, thus leading to hypoxia, while their further expansion makes them exceed the limits of oxygen diffusion. The increased mechanical and ischemic stress triggers inflammation and necrosis.

**Figure 2 ijms-22-13167-f002:**
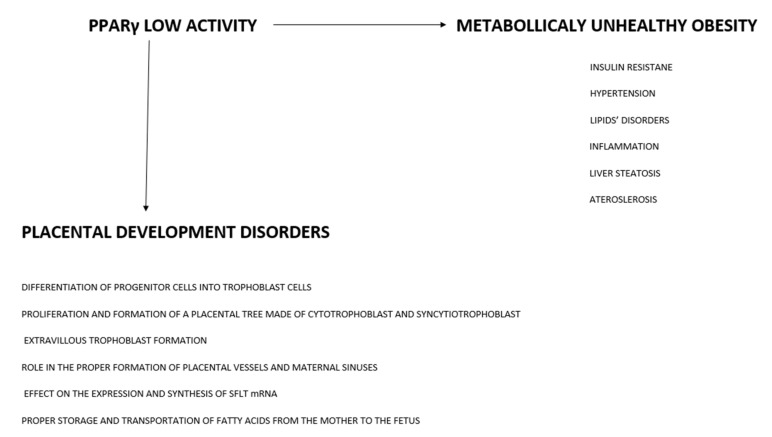
Diagram showing PPARγ—a factor linking metabolically unhealthy obesity with placental pathologies.

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
