# Peer review of "PPARγ—A Factor Linking Metabolically Unhealthy Obesity with Placental Pathologies"

_ijms, 2021, doi:10.3390/ijms222313167_

Round 1

Reviewer 1 Report

This manuscript is a review of the links between preeclampsia, obesity and PPAR gamma. The references cited are up-to-date and relevant. The text could be better organised into sub-paragraphs. Figure 1 should be revised for more clarity and efficiency, as it no longer corresponds to current standards. The manuscript would benefit from one or two additional figures presenting graphical summaries of the sometimes very dense content.
Thge differences between rodent and human placenta / placentation should be briefly discussed as a lot of studies referenced in the manuscript used mouse or rat models.

Author Response

Dear Reviewer,

Thank you very much for critical review of our article “ PPARγ – a factor linking metabolically unhealthy obesity with placental pathologies”. We have carefully considered your suggestions and have revised our manuscript accordingly; we hope that these changes meet with your approval.

Kind regards,

Ewa Kwiatkowska

Reviewer 2 Report

Kwiatkowski et al. have described in their review that "PPARγ – a factor linking metabolically unhealthy obesity with placental pathologies". The manuscript discusses the pathologies of distinct pregnancy disorder preeclampsia and a possible contribution of maternal obesity to preeclampsia. My concerns are highlighted in the following points.

  1. Maternal obesity is increasing globally. It also contributes to gestational diabetes mellitus. The authors have failed to address this in their review.
  2. What is the rationale for focusing specifically on preeclampsia.
  3. What are the underlying biological mechanisms that are common for preeclampsia and maternal obesity. 
  4. PPARg and its role in placental development, signalling and its association with pregnancy pathologies are well known in the field. However, it is not clear how the authors propose to identify a novel mechanism to link maternal obesity to preeclampsia specifically is unclear.

Author Response

(The authors gave the same response as above.)

Round 2

Reviewer 2 Report

There are no further comments to add.